# Evaluation of Five Buffers for Inactivation of Monkeypox Virus and Feasibility of Virus Detection Using the Panther Fusion^®^ Open Access System

**DOI:** 10.3390/v14102227

**Published:** 2022-10-10

**Authors:** Robert J. Fischer, Shane Gallogly, Jonathan E. Schulz, Neeltje van Doremalen, Vincent Munster, Sanchita Das

**Affiliations:** 1Laboratory of Virology, National Institute of Allergy and Infectious Diseases, National Institutes of Health, Hamilton, MT 59840, USA; 2Department of Laboratory Medicine, Clinical Center, National Institutes of Health, Bethesda, MD 20892, USA

**Keywords:** monkeypox virus, inactivation, molecular detection, automated PCR

## Abstract

Rapid diagnosis is key to containing viral outbreaks. However, for the current monkeypox outbreak the major deterrent to rapid testing is the requirement for higher biocontainment of potentially infectious monkeypox virus specimens. The current CDC guidelines require the DNA extraction process before PCR amplification to be performed under biosafety level 3 unless vaccinated personnel are performing assays. This increases the turn-around time and makes certain laboratories insufficiently equipped to handle specimens from patients with suspected monkeypox infection. We investigated the ability of five commercially available lysis buffers and heat for inactivation of monkeypox virus. We also optimized the use of monkeypox virus in Hologic^®^ Panther Specimen Lysis Buffer for detection of virus in the Panther Fusion^®^ Open Access System using published generic and clade specific monkeypox virus primers and probes.

## 1. Introduction

Since May 2022, the largest number of monkeypox virus (MPXV) cases have been described from non-endemic countries. The World Health Organization acknowledges the emergent nature of the event and recommends surveillance and rapid diagnosis as a strategy to control the outbreak. The optimum diagnosis of MPXV relies on molecular testing by PCR of specimens collected directly from a skin lesion. Ideally, CDC recommends collection of two swabs each from multiple lesions, it is preferred that the swabs are collected from lesions in different locations on the body and with differing appearances, it is also recommended that the lesions be adequately sampled [1]. The current CDC non-variola orthopoxvirus assay is FDA cleared for PCR based detection of MPXV virus from lesions. However, manipulation of specimens for DNA extraction prior to PCR testing needs to be performed in facilities that either have recently vaccinated personnel or advanced biosafety precautions (https://www.cdc.gov/poxvirus/monkeypox/lab-personnel/lab-procedures.htmL) (accessed on 1 September 2022). Laboratories will therefore need to be attentive to biosafety procedures that may significantly delay diagnostic testing. The procedures used for extraction of nucleic acid from clinical material have been shown to inactivate viral pathogens in previous studies [2,3]. However, studies on Ebola virus have shown that not all lysis buffers are adequately effective in inactivating viruses in some clinical specimens when used alone [4,5]. Similarly, Pastorino et al. have demonstrated that not all lysis buffers were equally effective in inactivating SARS-CoV-2 [6]. Therefore, concern for safety among laboratory personnel remain when performing diagnostic assays. Additionally, manipulation of specimens under stringent biosafety conditions pose a bottleneck for high throughput testing which is the key to rapid response and control of the current MPXV outbreak. Several high throughput automated platforms have an open access functionality that can be used to adapt existing real-time PCR primers and probes for an automated testing further reducing manipulation of specimens and increasing general efficiency in the diagnostic laboratory for rapid detection of MPXV. The real-time PCR assay developed by Li et al. [7] includes a generic primer probe that detects MPXV and separate primer probe combinations to differentiate MPXV Clade I and Clade II. We adapted the generic MPXV and the Clade II specific primer probes for detection of Clade II MPXV on the Panther Fusion^®^ Open Access (Fusion) platform (Hologic Inc. San Diego, CA, USA). The Fusion platform has been used extensively for SARS-CoV-2 and respiratory virus detection in routine clinical laboratory [8], the adaptation of a MPXV assay would be useful for the automated clinical laboratory workflow.

In this study we evaluated lysis buffers commonly used in clinical laboratories to test their ability to inactivate MPXV, including commercial specimen transport medium to facilitate rapid testing for MPXV in clinical specimens. In addition, we report the feasibility of using the Fusion for rapid automated detection of MPXV using generic and clade specific primers and probes adapted from Li et al. [7].

## 2. Methods

### 2.1. Procedure for Sample Processing to Evaluate Viral Inactivation

Stock preparation of MPXV were used for the evaluation of inactivation. Briefly, MPXV was incubated with the selected inactivation reagent or at the desired time for heat inactivation. To remove reagents, samples were subjected to desalting, detergent removal or filtration centrifugation. Inactivation was evaluated by plaque assay, where inactivation was scored as the absence of plaque formation by MPXV.2.2. Buffer AVL (Qiagen), Trizol™, and Panther Fusion Specimen Lysis reagents

To test MPXV virus inactivation each reagent was tested in triplicate with an additional no virus control to ensure that the reagent removal method sufficiently removed enough reagent to prevent cell cytotoxicity in the plaque assay. The virus stock used was hMPXV/USA/MA001/2022 with a viral titer of 4.8 × 10^6^ plaque forming units (PFU)/mL (corresponding Ct value 9.04). Samples were processed with Pierce^®^ Detergent removal Spin Columns (4 mL), Pierce^®^ Zeba™ Spin Desalting Columns and or Amicon^®^ Ultra-15 30 K centrifugal filter units (Merk Millipore). Panther Fusion Urine Transport Medium (UTM) was additionally processed using 2 passes through Pierce^®^ Detergent removal Spin Columns (4 mL) and using an Amicon^®^ Ultra-15 30 K centrifugal filter units. Trizol™ was additionally processed with Amicon^®^ Ultra-15 30 K centrifugal filter units (see Table 1).

### 2.2. Sample Inactivation with Buffer AVL

Buffer AVL (Qiagen) inactivation was performed following the manufacturer’s instructions (see Table 1). For each reagent removal method used, 140 uL of stock virus was added to each of 3 vials containing 560 µL of AVL. In one control vial 140 uL of Dulbecco’s Modified Eagle Medium (Sigma) supplemented with 2% fetal bovine serum (FBS, Gibco), 100 U/mL penicillin and 100 µg/mL streptomycin ((Gibco) (2% DMEM)), was added to a control vial containing 560 µL of AVL. These were incubated for 10 min at room temperature, after which the contents of the vial were transferred into 560 uL of absolute ethanol in a second vial and incubated at room temperature for an additional 10 min, followed by the reagent removal column.

### 2.3. Sample Inactivation with Trizol™

Trizol™ inactivation was performed following the manufacturer’s instructions (see Table 1). 250 uL of stock virus was added to each of 3 vials containing 750 µL of Trizol™, 250 uL of 2% DMEM was added to a control vial containing 750 µL of Trizol™. These were incubated for 10 min at room temperature followed by the reagent removal column.

### 2.4. Sample Inactivation with Panther Fusion Specimen Transfer Medium (STM), Blood Transfer Medium (BTM) and Urine Transfer Medium (UTM)

STM, BTM, and UTM inactivations were performed following the manufacturer’s instructions. For each, the volume of stock virus from Table 1 was added to each of 3 vials containing the corresponding volume of reagent from Table 1 and for the controls 2% DMEM was substituted for viral stock and added to the appropriate volume of reagent. These were incubated for 10 min at room temperature, followed by the reagent removal column.

### 2.5. Sample Inactivation with Heat Treatment

Three 2 mL vials containing 1 mL of stock virus were placed into an Eppendorf Thermomixer 2.0. The wells of the thermomixer were partially filled with 1 mL of water to provide consistent heat transfer to the contents of the vials. The vials were kept at temperature for the times indicated in Table 1. After removing the vials from the heat, they were floated in cold tap water for 10 min to cool them prior to adding to the cell plates. The total content of each vial was added to a well of a 6-well plate and then treated as described in the plaque assay section below.

### 2.6. Reagent Removal

In an attempt to eliminate any cell cytotoxicity due to the inactivation reagent, a reagent removal step was performed for each sample. All samples were subjected to a reagent removal step using either Pierce^®^ Detergent removal Spin Columns (4 mL) or Pierce^®^ Zeba™ Spin Desalting Columns or Amicon ^®^ Ultra-15 30 K Centrifugal Filters.

*Pierce^®^ Detergent removal Spin Columns (4 mL):* The detergent removal columns were prepared according to the manufacturer’s instructions. To remove the active lysis agents from the sample 1 mL of the sample was slowly applied to the center of the columns resin bed and incubated at room temperature for 2 min. The column was spun at 1000× *g* for 2 min. The volume of the collected sample was brought up to 3 mL for addition to cells in a 6-well plate.

*Pierce^®^ Zeba™ Spin Desalting Columns (2 mL):* The desalting columns were prepared according to the manufacturer’s instructions. To remove the active lysis agents 700 µL of the sample was slowly applied to the center of the columns resin bed. After the sample had absorbed into the resin 40 µL of ultrapure water was added on top of the resin. The column was spun at 1000× *g* for 2 min. The volume of the collected sample was brought up to 3 mL for addition to cells in a 6-well plate.

*Amicon^®^ Ultra-15 30 K centrifugal filter units:* The samples were treated following the manufacturer’s instructions. Briefly the samples were added to the column chamber in their entirety and centrifuged at 4000× *g* for 10 min. The chambers were then rinsed with 1 mL PBS and spun again at 4000× *g* for 10 min. Virus was recovered by adding 500 µL of 2% DMEM to the chamber, being sure to rinse the sides of the chamber, the resulting rinse was transferred to a new tube and 2% DMEM added to bring the final volume up to 3 mL for addition to cells in a 6-well plate.

To overcome the incompatibility of the high phenol content of Trizol™ with the Amicon^®^ filters the Trizol™ samples were diluted 1:30 with PBS and split between two Amicon^®^ filters. Each filter was rinsed with 500 µL of PBS and the two samples combined. The resulting 1 mL of sample was brought up to 3 mL with 2% DMEM and treated as below.

### 2.7. Virus Recovery from Specimens following Reagent Removal Procedures

To ensure that a negative result was not due to virus loss during the reagent removal procedure a sample consisting of 500 µL of virus stock and 500 µL of 2% DMEM was treated with each of the reagent removal procedures described above.

*Plaque assay:* The cell culture medium was removed from 6-well plates containing confluent Vero E6 cells (a kind gift from Professor Ralph Baric, UNC) and replaced with 1 mL of sample/well, in triplicate. The plates were incubated at 37 °C and 5% CO_2_ for 2 h then and an additional 1 mL of 2% DMEM was added to each well. After 3 days the growth medium was removed from each well and replaced with 500 µL of 10% formalin and incubated at room temperature for 10 min. The formalin was removed and 500 µL of 1% crystal violet (Sigma, V5265) was added to each well and incubated at room temperature for 10 min. Crystal violet was removed, and wells were washed with tap water. Plaques were counted in each well.

### 2.8. Panther Fusion Open Access Platform for MPX Assay Optimization

We used MPXV generic and clade II specific primers and Taqman probes described by Li et al. [6] on the fully automated Panther Fusion^®^ Open Access (Fusion) platform. The generic and clade specific assays were run as separate reaction as both probes were FAM labelled. The internal control primer and probes were the proprietary available for Fusion assays and was multiplexed with both assays. The primers and probe sequences and concentrations used are detailed in Table 2. The concentrations of KCl, MgCl_2_ and Tris buffer were adjusted for conditions optimal for use on Fusion platform. The PCR cycling conditions were also optimized and for both generic and clade specific assays were 1 cycle at 95 °C followed by 45 cycles of 95 °C for 8 s with 60 °C for 25 s, the reagent cartridges containing enzyme and nucleotides were obtained from Hologic and used according to manufacturer’s instructions.

*Material Used for Assay Optimization*. The following viral control material was used for assay optimization. Irradiated MPXV strains hMPXV-USA-2003-039 ^6^, hMPXV/USA/MA001/2022 and hMPXV/USA/FL002/2022. The irradiated viral strains were spiked into Universal Viral Transport Medium (VTM) to a final specimen volume of 500 µL. These control materials were then used as specimens and added to Hologic STM per manufacturer’s instruction (500 µL of specimen in a volume of 0.71 mL of buffer containing tube).

Contrived clinical material was generated by spiking irradiated MPXV at concentrations ranging from of 2.4 × 10^5^ to 4.8 × 10^4^ PFU/mL into swabs from skin lesions collected in VTM, these specimens had tested negative for viruses including herpes simplex virus-1 (HSV-1), herpes simplex virus-2 (HSV-2) and varicella zoster virus (VZV) (Table 3). We tested a total of five contrived clinical specimens. We also included eleven specimens from lesions that were positive for monkeypox at a reference laboratory, for testing specificity eight specimens were included that were negative for all viruses. For clinical specimens from lesions, 500 µL of the VTM that the swab was collected in, was added to STM buffer in a BSL-3 laboratory and held for 10 min before testing on the Panther Fusion Open Access platform.

For determination of limit of detection of the assay on the Open Access platform, we used quantitative synthetic MPXV DNA obtained from ATCC (ATCC VR-3270SD). Synthetic material was diluted in molecular grade water. Ten-fold serial dilutions of the DNA were made starting at 2.8 × 10^3^ copies/µL to 0.28 copies/µL. We added 500 µL of each dilution to Hologic STM and run on the Fusion platform using parameters described above for both the generic and clade specific MPXV virus assay.

## 3. Results and Discussion

### 3.1. Verification of Inactivated Viruses in Plague Assay

Using the described methods treating MPXV with the following reagents Panther STM, Panther BTM, Trizol™, AVL with ethanol, and heat treatment under the following conditions: 30 min at 65 °C, 15 min at 65 °C and 15 min at 95 °C resulted in no plaque formation in a plaque assay (Table 4). The three reagent removal methods were tested to determine virus recovery from the reagent removal process. The two column reagent removal methods both had an 89% virus recovery when compared to untreated stock virus and the Amicon^®^ filter had no loss of virus. We used a high concentration of virus (10^6^ pfu/mL) to test the efficacy of the buffers in their ability to inactivate virus. We also used three different methods for reagent removal to make sure that the cytotoxicity of the lysis buffers did not interfere with the sensitivity of the viral culture post inactivation. Our results indicate that STM, BTM and heat were successful in virus inactivation.

Inactivation of MPXV following incubation for 10 min in the Panther UTM could not be confirmed as the results were inconclusive. Each of the reagent removal methods investigated resulted in cytotoxicity with with the cells mostly gone from the plate on day four after inoculation of the wells. This also held true for the negative control wells indicating that this was caused by incomplete removal of the active ingredient of the reagent rather than any cytopathic effect of the virus (Appendix A). Whereas the positive control wells did not show cytotoxicity. While for Trizol, the detergent and de-salting method were inconclusive, using an Amicon^®^ filter for reagent removal indicate that it was able to inactivate the virus confirming results of the UK Health security agency. https://www.gov.uk/government/publications/monkeypox-ukhsa-laboratory-assessments-of-inactivation-methods (accessed on 1 September 2022). Limited information is available on the relationship of the infectious viral load observed in samples vs. the dose we used in our experiment [9,10]. However, information is available on the Ct values of human samples, and the highest viral load is observed in samples obtained from lesions, with Ct values around 12–16, the Ct value of the virus used for inactivation was 9, suggesting that the viral load we used is higher than what is typically observed in human lesion samples. Further validation experiments maybe required to ascertain the inactivation efficiency of these reagents using clinical specimens, since biological matrix may influence the optimum efficacy of lytic activity [2].

### 3.2. PCR Method Optimization

The generic and clade specific MPXV assays were able to detect all control materials tested. The contrived clinical specimens were positive at a Ct value ranging from 24–32. We also tested 11 clinical known positive skin lesion specimens and they all tested positive with Ct values between 19.7 and 30.0 (for generic assay) and 20.5 and 30.4 (for clade specific assay) (Table 3). Additionally, specificity of the assay was ascertained using eight known clinical specimens (skin lesions and mucosal swabs) and tested negative by both assays. We used Panther Fusion proprietary internal control (spiked in within the instrument during specimen extraction) and a positive result from the internal control was required for a valid result for the MPXV PCR assay. All results with Ct values < 37 were considered positive based on our results from the limit of detection experiments as described below. Since the total number of confirmed clinical MPXV positive specimens were low in the assay validation, we incorporated additional contrived specimens to ascertain how the assay would work on matrix most clinically relevant (i.e., skin swabs and mucosal lesions).

### 3.3. Detection Limit of Panther Fusion Open Access Platform

Using synthetic material, the limit of detection for both assays were at 2.8 DNA copies/µL (Figure 1). The dilution containing 0.28 copies/µL was positive but with Ct values ≥ 37 and was not consistently detected. Similarly, further dilutions of synthetic DNA with copy numbers between 2.8 and 0.28 did not yield consistent Ct values when performed over time, therefore these results were considered inconclusive setting the limit of detection of the assay at 2.8 copies/µL. Recent publication has shown that lower Ct values may be reflective of false positive results and PCR results from specimens with Ct values > 34 should be carefully evaluated [11]. This agrees with our finding of inconsistent Ct values in lower dilutions tested.

Our results suggest that specimens collected in STM can be used to test for MPXV on the Panther Fusion Open Access platform.

While most laboratory protocols assume that the lysis buffer in commercial nucleic acid extraction and amplification kits are able to inactivate viruses, there are several issues that could affect this assumption including the viral load in the specimen and the biological matrix being tested. We show that incubation for 10 min in two commercially available specimen collection buffers (STM and BTM) are effective in inactivating MPXV at a high concentration of 0.48 × 10^7^. While it is postulated that MPXV viral loads are higher in skin lesions than in pharyngeal swabs, these are extrapolated from Ct values obtained when samples were tested by PCR [12]. There are limited studies showing correlation between Ct value and MPXV viral load in clinical material and these show that the Ct values over 33 have viral copy numbers near the limit of detection (25 pfu/mL) [10]. Previous animal studies also have shown that culture is successful in lesions with viral load of 10^4^ to 10^5^ copies/mL [13]. We tested viral cultures with a high titer reflective of that seen in specimen collected from a MPXV lesion, our results suggest that specimens collected in STM, BTM, Trizol or AVL with ethanol renders the virus inactive thus enabling DNA extraction in non-BSL-3 facilities although this finding needs to be confirmed with clinical specimens. 

In this study we also investigated the feasibility of using the Panther Fusion Open Access platform as an automated system to test for MPXV using a modification of primer probe combinations described by Li et al. [7]. Our results show that this platform can be used to test specimens collected in STM or BTM tubes, thus reducing manipulation of specimens by laboratory personnel and reducing biohazard associated with testing of potential MPXV containing specimens.

Our study is limited in that we did not test for viral inactivation in biological specimens where the variable viral load and different matrices can affect the outcome of contact with lysis buffer. A study by van Kampen et al. have shown that viral inactivation of hemorrhagic fever viruses by detergents is annulled by the presence of serum, and organic load has long been known to affect the action of biocides on surfaces [14]. The presence of high protein can resist even heat inactivation as shown in previous studies using heat inactivation for vaccine production [15], as well the exact procedure for heat inactivation [16]. Therefore, these results should be confirmed for complex matrices with high protein content such as whole blood and serum. However, specimens from skin lesions are low on organic and cellular material and our results indicate that the buffers tested can inactivate MPXV virus at high concentrations.

A second limitation is that we were able to test very limited number of clinical specimens using this primer probe combinations and were unable to perform a complete clinical validation. Additionally, we did not use BTM collection buffer for assay optimization. However, similar primers (CDC non-variola orthopox virus assay) has been approved by the FDA and are being used in several clinical laboratories for detection of MPXV. Further, the purpose of this study was to demonstrate the feasibility of using Fusion Open Access Platform for MPXV assays and the use of STM which is shown to inactivate the virus thus improving the workflow in the clinical laboratory.

Additionally, current guidelines recommend testing of specimens from lesions which are more likely to be collected in STM than BTM, the latter being used for collection of blood. Our results also indicate that viral nucleic acid is stable and can be detected from specimens collected in STM. Our results in contrived specimens using matrix that is similar to that from MPXV lesion show that these primers are able to detect the virus at concentrations as low as 2.8 copies/µL. In addition, results from our study show that using STM for collection, the assay can be performed on an automated system thus making high throughput testing possible without the requirement of higher biosafety levels.

## 4. Conclusions

Commercially available buffers (excluding Panther UTM) have the ability to inactivate MPXV after 10 min of incubation. Additionally, previously described MPXV primers and probes are adaptable on the Panther Fusion^®^ Open Access automated platform. This indicates that clinical specimens collected in these buffers have the potential of being suitable for testing in laboratories in an automated platform without additional manipulation.

## Figures and Tables

**Figure 1 viruses-14-02227-f001:**
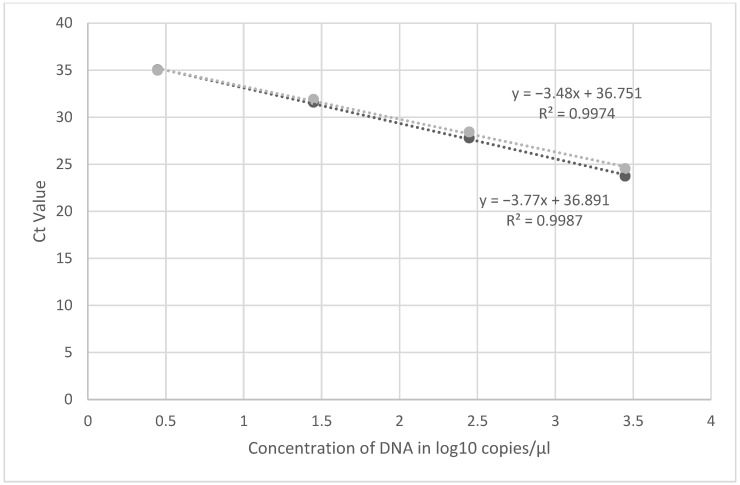
Linearity of the primer probe set (D9-5 and D9-6) using dilutions of synthetic MPXV DNA. Experiments were run in duplicate, mean Ct value was used for linearity.

**Table 1 viruses-14-02227-t001:** Conditions of MPXV inactivation trials for 8 treatment methods and the reagent removal methods employed to enable verification of inactivation by plaque assay.

	Inactivation Method					
Reagent	Test Volumes Used (Performed in Triplicate)		Control Volumes Used		Time	Reagent Removal Method
	Reagent Volume	Virus Volume	Reagent Volume	DMEM Volume		
Hologic^®^ Panther STM	710 µL	500 µL	710 µL	500 µL	10 min	de-salt
	710 µL	500 µL	710 µL	500 µL		detergent
Hologic^®^ Panther BTM	1200 µL	400 µL	1200 µL	400 µL	10 min	de-salt
	1200 µL	400 µL	1200 µL	400 µL		detergent
Hologic ^®^ Panther UTM	950 µL	500 µL	950 µL	500 µL	10 min	de-salt
	950 µL	500 µL	950 µL	500 µL		detergent
	950 µL	500 µL	950 µL	500 µL		detergent 2X
	950 µL	500 µL	950 µL	500 µL		Amicon filter
Trizol™	750 µL	250 µL	750 µL	250 µL	10 min	de-salt
	750 µL	250 µL	750 µL	250 µL		detergent
	750 µL	25 min 0 µL	750 µL	250 µL		Amicon filter
Buffer AVL & Ethanol §	560 µL	140 µL	560 µL	140 µL	10 min & 10 min	de-salt
	560 µL	140 µL	560 µL	140 µL		detergent
Heat Treatment						
65 °C	1 mL	1 mL			30 min	
65 °C	1 mL	1 mL			15 min	
95 °C	1 mL	1 mL			15 min	
Virus Control	500 µL ^#^	500 µL				de-salt
	500 µL ^#^	500 µL				detergent
	500 µL ^#^	500 µL				Amicon filter
	500 µL ^#^	500 µL				No treatment control

§ Volume of ethanol used was 560 µL and incubation time was 10 min. ^#^ DMEM was used as reagent for virus control. DMEM: Dulbecco’s minimum essential medium.

**Table 2 viruses-14-02227-t002:** Oligonucleotides used in this study.

Assay Name	Primer/Probe Oligonucleotide Sequence (5′-3′)	Concentration (µM)
Generic Monkeypox virus		
D9-5F_MPXV_Uni	GGAAARTGTAAAGACAACGAATACAG	0.3
D9-5F_MPXV_Uni	GCTATCACATAATCTGGAAGCGTA	0.3
D9-5P_MPXV_Uni_	FAM- AAGCCGTAATCTATGTTGTCTATCGTGTCC-ZEN_IBHQ	0.2
Clade specific Monkeypox virus		
D9-6F_MPXV_WA_	CACACCGTCTCTTCCACAGA	0.3
D9-6R_MPXV_WA_	GATACAGGTTAATTTCCACATCG	0.3
D9-6P_MPXV_WA_	FAM-AACCCGTCGTAACCAGCAATACATTT-ZEN_IBHQ	0.2
Internal Control		
Universal IC	Proprietary/Quasar705	0.6/0.4

**Table 3 viruses-14-02227-t003:** Details and results obtained from contrived and clinical MPXV specimens.

Specimen Number	Specimen Type (Swab Collected in VTM)	Irradiated Viral Strain Used to Spike (Concentration of Virus Used)	Ct Values Obtained on Fusion *	
			D9-5 generic MPXV assay	D9-6 clade specific MPXV assay
S1 (Contrived)	Mouth mucosal lesion	hMPXV/USA/MA001/2022 (2.4 × 10^5^)	25.7	26.1
S2 (Contrived)	Skin lesion	hMPXV/USA/FL002/2022 (2.4 × 10^5^)	26.8	27.3
S3 (Contrived)	Skin lesion	hMPXV/USA/MA001/2022 (4.8 × 10^4^)	28.0	28.3
S4 (Contrived)	Skin lesion	hMPXV-USA-2003-039 (2.4 × 10^5^)	22.8	23.5
S5 (Contrived)	Skin lesion	hMPXV/USA/FL002/2022 (4.8 × 10^4^)	25.3	26.6
S6 (Clinical)	Skin lesion (confirmed positive for MPXV)	NA (clinical positive)	28.5	29.2
S7 (Clinical)	Skin lesion (confirmed positive for MPXV)	NA (clinical positive)	21.2	22.2
S8 (Clinical)	Skin lesion (confirmed positive for MPXV)	NA (clinical positive)	30.0	30.4
S9 (Clinical)	Skin lesion (confirmed positive for MPXV)	NA (clinical positive)	22.6	23.2
S10 (Clinical)	Skin lesion (confirmed positive for MPXV)	NA (clinical positive)	23.0	23.5
S11 (Clinical)	Skin lesion (confirmed positive for MPXV)	NA (clinical positive)	19.7	20.5
S12 (Clinical)	Skin lesion (confirmed positive for MPXV)	NA (clinical positive)	19.9	20.7
S13 (Clinical)	Skin lesion (confirmed positive for MPXV)	NA (clinical positive)	23.1	23.4
S14 (Clinical)	Skin lesion (confirmed positive for MPXV)	NA (clinical positive)	26.0	26.5
S15 (Clinical)	Skin lesion (confirmed positive for MPXV)	NA (clinical positive)	24.1	24.6
S16 (Clinical)	Skin lesion (confirmed positive for MPXV)	NA (clinical positive)	28.7	29.1

* Indicates results obtained on the Panther Fusion^®^ Open Access platform using generic and clade specific MPXV assays.

**Table 4 viruses-14-02227-t004:** Results of MPXV inactivation testing using various methods and three reagent removal methods.

Reagent	Time	Reagent Removal Method	Plaque Count								
			Sample 1			Sample 2			Sample 3		
			Well 1	Well 2	Well 3	Well 1	Well 2	Well 3	Well 1	Well 2	Well 3
Panther STM	10	detergent	0	0	0	0	0	0	0	0	0
Panther BTM	10	de-salt	0	0	0	0	0	0	0	0	0
		detergent	0	0	0	0	0	0	0	0	0
Trizol	10	de-salt	0	0	0	0	0	0	0	0	0
		Amicon	0	0	0	0	0	0	0	0	0
AVL & EtOH	10	detergent	0	0	0	0	0	0	0	0	0
	Time	Temperature	Sample 1	Sample 2	Sample 3						
Heat	30 min	65	0	0	0						
	15 min	65	0	0	0						
	15 min	95	0	0	0						
			Plaques	Dilution	PFU						
Virus loss controls (titrations)		untreated stock	9	5	9 × 10^5^						
		de-salt	8	5	8 × 10^5^						
		detergent	8	5	8 × 10^5^						
		Amicon	16	5	1.6 × 10^6^						

## Data Availability

Data is included in the manuscript. No additional data has been reported for this study.

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
