# Peer review of "Evaluation of Five Buffers for Inactivation of Monkeypox Virus and Feasibility of Virus Detection Using the Panther Fusion^®^ Open Access System"

_viruses, 2022, doi:10.3390/v14102227_

Round 1

Reviewer 1 Report

A sound study.

I have little doubt that the concentration of virus tested in the manuscript represents a high dose but I believe there would be value in a brief survey of the literature to confirm this. This would bolster confidence that the conditions tested were effective at inactivating levels of virus expected (and above) found in authentic samples.

Author Response

We would like to thank the reviewer for the comment. We have included a paragraph (lines 253-259 of the revised manuscript) and also cited two recent references that detail viral load and Ct values to correlate the use of  4.8 x 106 pfu/ml for the inactivation experiment. We also added a corresponding Ct value for this viral load in line 79.

Reviewer 2 Report

Evaluation of Five Buffers for Inactivation of Monkeypox Virus 2 and Feasibility of Virus Detection Using the Panther Fusion® 3 Open Access System

The manuscript addresses the important issue of working with MPXV at containment and seeks to test inactivation protocols to allow downgrading the containment required to BSL2 for Real-time PCR testing, in particular for the use of the Panther Fusion® Open Access automated platform.

This is a well written manuscript with only a few minor faults. I would recommend publication after addressing the comments below.

Line 38: the authors may also like to include SARSCoV2 within this statement Pastorino et al (doi.org/10.3390/v12060624)

Methods section seems to have lost some subsection numbers

Line 98: should this be a new section 2.6?

Line 131: should this be 2.7?

Line 140 should this be 2.8?

Line 146 should be KCl

Line 192: This is confusing to me – perhaps I am being a bit dense? Following replacement of the media in plates with the same samples that underwent plaque assay, at 4 days pi the cell sheets were destroyed indicating toxicity carryover. The same control samples were plaqued giving plaques at -5 dilution – did the -1 and -2 dilutions also cause the cell sheet disruption? Perhaps this could be clarified?

Line 209: The sentence “Using synthetic material, the limit of detection for both assays were at 2.8 DNA 209 copies/μl using the (Figure 1).” does not read correctly.

Line 212: The link provided did not work

Would have been nice to have included more concentrations of DNA at the lower end for greater definition of the lower limit of detection (to give a Ct value between 35 and 37).

Line 226: There has been correlation between viral load in clinical material and Ct: Paran et al, Euro Surveill. 2022;27(35):pii=2200636. https://doi.org/10.2807/1560-7917.ES.2022.27.35.2200636

Author Response

The authors would like to thank the reviewer for thorough evaluation and suggestions, we appreciate this opportunity provided to us to improve this manuscript. Here is a point by point response to the comments:

Line 38: the authors may also like to include SARSCoV2 within this statement Pastorino et al (doi.org/10.3390/v12060624):

Response: This reference has been added and we included a comment regarding the inactivation of SARS-CoV-2 (lines 43-45 of the revised manuscript).

Methods section seems to have lost some subsection numbers:

Response: We apologize for this error in organization of the subsections. This has been corrected in the revised manuscript. 

Line 98: should this be a new section 2.6?

Response: This is a new section in the revised manuscript

Line 131: should this be 2.7?

Response: This is a new section in the revised manuscript

Line 140 should this be 2.8?

Response: This is a new section in the revised manuscript

Line 146 should be KCl:

Response: We apologize for this typographical error and have corrected this in the revised manuscript. 

Line 192: This is confusing to me – perhaps I am being a bit dense? Following replacement of the media in plates with the same samples that underwent plaque assay, at 4 days pi the cell sheets were destroyed indicating toxicity carryover. The same control samples were plaqued giving plaques at -5 dilution – did the -1 and -2 dilutions also cause the cell sheet disruption? Perhaps this could be clarified? 

Response: Yes, that is correct, with the positive control samples, although they were at -5; the -1 and -1 showed clear plaques and not cytotoxicity. We apologize for the lack of clarity of this part of the results and have incorporated revisions to clarify in lines 244-249 of the revised manuscript. 

Line 209: The sentence “Using synthetic material, the limit of detection for both assays were at 2.8 DNA 209 copies/μl using the (Figure 1).” does not read correctly.

Response: This was a typographical error, we have corrected this in the revised manuscript. 

Line 212: The link provided did not work.

Response: We apologize for this confusion, since the preparation of this manuscript, there has been some changes in this document. Based on new evidence published in MMWR Morb Mortal Wkly Rep 2022, 71 (36), 1155-1158. DOI: 10.15585/mmwr.mm7136e1, we revised this section of the manuscript. Lines 283-286 and removed the link to the document that did not work.

Would have been nice to have included more concentrations of DNA at the lower end for greater definition of the lower limit of detection (to give a Ct value between 35 and 37). 

Response: We attempted testing lower dilutions but did not get consistent Ct values <37 in dilutions below 2.8 copies/ul. Also, with the recent publication showing the unreliability of Ct values >34 we revised the discussion section (lines 284-286).

Line 226: There has been correlation between viral load in clinical material and Ct: Paran et al, Euro Surveill. 2022;27(35):pii=2200636. https://doi.org/10.2807/1560-7917.ES.2022.27.35.2200636.

Response: We thank the reviewer for directing us to this publication, we have included this in the revised manuscript (lines 302-304).

Reviewer 3 Report

Manuscript viruses-1960626 by Fischer et al. describes the efficiency of different lysis buffers on monkeypox virus (MPXV) inactivation. Authors evaluate the possible viruses that might survive lysis buffers in plague assay, optimize PCR method, and prove the concept of applying automated Panther Fusion Open Access platform into virus infected sample analysis. This would protect lab personnel from getting infected and allow high throughput testing.

This is a well designed paper which is highly valuable for rapid testing of MPXV and its prevention, but I recommend to reorganize the 'Method' section for better clarity. Please see the following comments:

Introduction: it is recommended that authors include descriptions about the MPXV generic and clade II primer and probe method developed by Li et al. and include introduction of the Panther Fusion Open Access platform, and its applications in other virus testing.

Section 2.1: it is recommended that authors briefly describe about the generic procedures of sample processing for verifying virus inactivation in plague assay. e.g. the generic procedures started with virus inactivation, followed by reagent removal, and ended with plague assay. Then described details in the subsequent sections. The subtitle for 2.1 should reflect this is generic procedures. 

Section 2.2-2.5: it is recommended that authors add 'Sample Inactivation with ' in the title of each subsection to emphasize the purpose of the method. e.g. in Section 2.2, instead of 'Buffer AVL', may consider use 'Sample Inactivation with Buffer AVL'.

Lines 98, 127, 131, 140, 152: it is recommended that authors add subsection in each core paragraphs in Section 'Method'. e.g. in Line 98, instead of 'Reagent Removal', may consider use '2.6. Reagent Removal'.

Table 1: it is recommended to put Table 1 behind inactivation methods and reagent removal methods; also it is recommended to move column 'Reagent Removal Method' to the last column, and then add a merged cell to include the rest of columns and named with 'Inactivation method' so that it's clear columns 'Reagent', 'Test Volumes Used', 'Control Volumes Used' and 'Time' are all related to the 'Inactivation method'. The time for 'Buffer AVL & Ethanol' should be '10 mins and 10 mins' since each procedure require 10 mins.  The 'Tested Volume Used' for 'Heat Treatment' should be '1 mL' instead of '1 mL' under 'Control Volume Used' as described in Line 91. The 'Tested Volume Used' for 'Virus Control' should be '500 uL  500 uL' instead of 'Control Volume Used' as described in Line 129.

Line 112: did you mean to add stacker of ultrapure water to ensure maximum sample recovery?

Line 161-162: it is recommended to include the full name of HSV-1, HSV-2 and VZV before introduce acronyms.

Results: it is recommended to include subsections and titles of subsections to emphasize the purpose of each experiment. e.g. Section 3.1 'Verification of inactivated viruses in plague assay', Section 3.2 'PCR method optimization', Section 3.3 'Detection limit of Panther Fusion Open Access platform'. For Section 3.2, it is recommended to discuss 'Table 3' in this section. What Ct values are defined as positive and why contrived samples and clinical samples with presented Ct values are considered as optimized condition.

Author Response

Introduction: it is recommended that authors include descriptions about the MPXV generic and clade II primer and probe method developed by Li et al. and include introduction of the Panther Fusion Open Access platform, and its applications in other virus testing.

Response: We thank the reviewer for this recommendation and feel that this strengthens our manuscript. We have included a paragraph and a citation for Fusion respiratory virus assay in the revised manuscript (lines 48-59)

Section 2.1: it is recommended that authors briefly describe about the generic procedures of sample processing for verifying virus inactivation in plague assay. e.g. the generic procedures started with virus inactivation, followed by reagent removal, and ended with plague assay. Then described details in the subsequent sections. The subtitle for 2.1 should reflect this is generic procedures.

Response: A new section 2.1 has been included in the revised manuscript.

Section 2.2-2.5: it is recommended that authors add 'Sample Inactivation with ' in the title of each subsection to emphasize the purpose of the method. e.g. in Section 2.2, instead of 'Buffer AVL', may consider use 'Sample Inactivation with Buffer AVL'.

Response: These recommendations have been incorporated in the revised manuscript.

Lines 98, 127, 131, 140, 152: it is recommended that authors add subsection in each core paragraphs in Section 'Method'. e.g. in Line 98, instead of 'Reagent Removal', may consider use '2.6. Reagent Removal'.

Response: We thank the reviewer and have incorporated these recommendations in the revised manuscript.

Table 1: it is recommended to put Table 1 behind inactivation methods and reagent removal methods; also it is recommended to move column 'Reagent Removal Method' to the last column, and then add a merged cell to include the rest of columns and named with 'Inactivation method' so that it's clear columns 'Reagent', 'Test Volumes Used', 'Control Volumes Used' and 'Time' are all related to the 'Inactivation method'. The time for 'Buffer AVL & Ethanol' should be '10 mins and 10 mins' since each procedure require 10 mins.  The 'Tested Volume Used' for 'Heat Treatment' should be '1 mL' instead of '1 mL' under 'Control Volume Used' as described in Line 91. The 'Tested Volume Used' for 'Virus Control' should be '500 uL  500 uL' instead of 'Control Volume Used' as described in Line 129.

Response: We thank the reviewer for suggesting these changes, a new Table 1 has been included and incorporated behind inactivation methods in the revised manuscript.

Line 112: did you mean to add stacker of ultrapure water to ensure maximum sample recovery?

Response: Yes, that was the purpose of this step.

Line 161-162: it is recommended to include the full name of HSV-1, HSV-2 and VZV before introduce acronyms.

Response: We apologize for this oversight and have corrected this in the revised manuscript.

Results: it is recommended to include subsections and titles of subsections to emphasize the purpose of each experiment. e.g. Section 3.1 'Verification of inactivated viruses in plague assay', Section 3.2 'PCR method optimization', Section 3.3 'Detection limit of Panther Fusion Open Access platform'. For Section 3.2, it is recommended to discuss 'Table 3' in this section. What Ct values are defined as positive and why contrived samples and clinical samples with presented Ct values are considered as optimized condition.

Response: We thank the reviewer for this suggestion and have incorporated this in the revised manuscript, we also included a section describing Table 3 in lines 273-280.